# Content Analysis of Multi-Annual Time Series of Flood-Related Twitter (X) Data

Nadja Veigel[1,2,3], Heidi Kreibich[2], Jens de Bruijn[4,5], Jeroen C.J.H. Aerts[5, 6], and Andrea Cominola[1,3]

[1]Chair of Smart Water Networks, Technische Universität Berlin, Straße des 17. Juni 135, Berlin, 10623, Germany
[2]Section 4.4 Hydrology, GFZ German Research Centre for Geosciences, Telegrafenberg, Potsdam, 14473, Germany
[3]Einstein Center Digital Future, Robert-Koch-Forum Wilhelmstraße 67, Berlin, 10117, Germany
[4]International Institute for Applied Systems Analysis (IIASA), Laxenburg, Austria
[5]Institute for Environmental Studies, VU University, Amsterdam, the Netherlands
[6]Deltares Institute, Delft, the Netherlands

**Correspondence:** Nadja Veigel (nadja.veigel@tu-berlin.de)

**Abstract.** Social media can provide insights into natural hazard events and people's emergency responses. In this study, we present a natural language processing analytic framework to extract and categorize information from of 43,287 textual Twitter (X) posts in German since 2014. We implement Bidirectional Encoder Representations from Transformers in combination with unsupervised clustering techniques (BERTopic) to automatically extract social media content, addressing transferability issues that arise from commonly used bag-of-word representations. We analyze the temporal evolution of topic patterns, reflecting behaviors and perceptions of citizens before, during, and after flood events. Topics related to low-impact riverine flooding contain descriptive hazard-related content, while the focus shifts to catastrophic impacts and responsibilities during high-impact events. Our analytical framework enables analyzing temporal dynamics of citizens' behaviors and perceptions which can facilitate lessons learned analyses and improve risk communication and management.

## 1 Introduction

Flood frequency and severity of impacts are exacerbated by climate change and urbanization (Paprotny et al., 2018). Developing new strategies to improve human response to flooding is crucial to safeguard lives, protect property, and enhance community resilience (Baldassarre et al., 2015).

Human response to natural hazards improves with their ability to communicate, share information, and experiences (Mileti, 1995; McCarthy et al., 2007; Giordano et al., 2017; Hong et al., 2018; Sermet and Demir, 2018). An emerging research topic is the role of social media in the communication of disaster risk management (Sermet and Demir, 2018; Zhang et al., 2019). Social media is used to quickly distribute critical information, enable real-time communication, aid in emergency response coordination, and provide a platform for affected individuals to share firsthand observations, insights, and personal experiences (Houston et al., 2015). Those mechanisms help enhance situational awareness, support, and resilience (Houston et al., 2015). For many years, individuals and organizations have engaged with social media platforms alongside traditional means of communication (Houston et al., 2015). This frequent usage of social media provides new opportunities for risk

assessment and management (Fraternali et al., 2012; Lin et al., 2016). Social media captures immediate personal experiences and emotional impacts that might be overlooked in conventional assessments but lacks the standardized methodology and detailed technical measurements found in traditional sources. Therefore, analyses of social media data should not be seen, but as complementary analyses that enhance traditional flood impact assessments by providing rapid situational awareness and capturing the social dimensions of flood impacts that might otherwise go undocumented.

Previous research has demonstrated correlations between the amount of tweets and hazard extent or impact (de Bruijn et al., 2019; Barker and Macleod, 2019; Sodoge et al., 2024). Furthermore, studies developed methodologies to evaluate the content (topics) and function of social media posts for specific hazard events (Kent and Jr., 2013; Cho et al., 2013; Huang and Xiao, 2015; Spence et al., 2015; Barker and Macleod, 2019; Donratanapat et al., 2020). Temporal and spatial patterns of social media use during disasters vary for different hazard types (Zhang et al., 2019). The rise of tweets related to floods or hurricanes is shallower and less abrupt than the spikes observed related to earthquakes (Cresci et al., 2017). Several case studies reported that users located close to a natural hazard, for example, the Horsethief Canyon Fire in 2012 (Kent and Jr., 2013) or Hurricane Sandy (Huang and Xiao, 2015), are more likely to post on social media than those at a distance. Huang and Xiao (2015) evaluated Twitter posts during Hurricane Sandy in 2012, showing that before the hurricane an increase in sharing traditional news outlets that published warnings was observed. During and after the event the tweets focused on reporting impact. During the 2011 earthquake in Japan Cho et al. (2013) assessed the content of tweets during a 40-hour period. They found that the tweets associated with emotional content decreased from 23.0% in the beginning to 5.3% in the aftermath of the earthquake. A study on Hurricane Sandy in 2012 revealed that, as the event unfolded, the number of tweets displaying emotional reactions increased, while those providing information about the hurricane decreased (Spence et al., 2015). Understanding the content of flood-related social media posts can be beneficial for risk management, but challenges related to social media data reliability and retrieving actionable information from social media (Gopal et al., 2024) are still open, along with the lack of long-term evidence on the effectiveness of crisis communication on social media (Lin et al., 2016). Furthermore, social media analyses can provide a basis for validating flood risk models based on reports and pictures of inundated areas and related impacts (Fohringer et al., 2015; Rözer et al., 2021).

While the literature consistently shows that it is feasible to deduct information on disaster risk and management from social media posts, the methodologies that are used to extract the contents lack transferability and the underlying data is mostly event-specific (Zhang et al., 2019; Gopal et al., 2024). Previously applied methodologies use a keyword-based pre-selection when retrieving content online and apply methodologies that rely on manual labels or word counts. Word meaning, frequency, and specific keywords change over time, making these approaches not adaptable to evolving language dynamics and new events. Additionally, the number of posts that can be analyzed is limited either by the availability of a labeled training data set, for example when using a supervised classification approach such as logistic regression (Huang and Xiao, 2015) or the feasibility of completely manual labeling (Cho et al., 2013; Spence et al., 2015). Another common approach is Latent Dirichlet Allocation (LDA) (Han and Wang, 2019; Aubert et al., 2013; Wu et al., 2021). Hierarchical Dirichlet Process (HDP) extends LDA by automatically determining the number of topics, enabling more flexible and scalable topic discovery. Latent Semantic Analysis (LSA) utilizes singular value decomposition to reduce dimensionality and capture underlying relationships between

terms and documents. Non-negative Matrix Factorization (NNMF) decomposes the term-document matrix into non-negative matrices (Churchill and Singh, 2022). However, since language and word usage can vary based on different events and places, these methods are not feasible for consistently studying multiple events. Moreover, word frequency based methods do not account for semantic relationships. Unsupervised approaches that do not require labeling and context dependent representation of the input data are required to apply content modeling over longer time spans automatically.

The recent development of open-source large-scale language models that are pre-trained on big text corpus data (see, e.g., Reimers and Gurevych (2019)) provide an opportunity to study multiple events, however, they are underrepresented in environmental modeling applications (Konya and Nematzadeh, 2024). Transformer models outperformed other embedding based content modeling approaches extracting information from textual Twitter (X) data on Covid-19 (Egger and Yu, 2022) and have been applied for sentiment analysis on geolocated tweets from Hurricane Ida (Tounsi et al., 2023). Based on these recent insights, the objective of our research is to analyse content of social media posts to gain knowledge about citizens' behaviour and perception of floods over a long time period for multiple, heterogeneous, flood events. In this study, we aim to develop a transferable approach for automatic extraction of content from multi-annual social media posts and to derive insights in the behaviors and perceptions of citizens before, during, and after flood events.

## 2 Materials and Methods

To track the content of flood-related textual Twitter (X) posts before, during, and after several flood events in Germany from 2014 to 2023, we employ a transformer-based model as our topic detection method. First, the text data is embedded into a high-dimensional vector space, leveraging the context-dependent meaning of the words contained. This approach ensures applicability across various events and large datasets in different languages. Next, utilizing the vectorized representation of the text data, we perform clustering to extract topics. The resulting clusters serve as a meaningful representation of the content in terms of topics within the data (Grootendorst, 2022).

To detect topics from tweets and categorize tweets in those topics we adapt the Topic Modelling pipeline proposed by Grootendorst (2022) to analyze contents and extract topics from flood-related textual Twitter (X) posts. Figure 1 shows the three main steps of our framework, which relies on textual Twitter (X) data as input: (1) a data preparation step where the input textual Twitter (X) data is prepared by cleaning, for example removing URLs, and filtering with non-flood related keywords (step *Data preparation and filtering* in Figure 1, Section 2.1, Supplementary Material, Section 2.1. (2) The *Content modeling - extracting topics from tweets* (Section 2.2) step is to extract a vectorized representation of the text (embeddings) utilizing a Sentence Transformer model (SBERT, version:paraphrase-multilingual-MiniLM-L12-v2, (Reimers and Gurevych, 2019)). Here, the text data is transformed, capturing the semantic meaning of sentences (box 2-a in Figure 1). This enables the model to understand the contextual relationships between words and phrases. To handle the high-dimensional nature of the embeddings, we apply a dimensionality reduction technique: Uniform Manifold Approximation and Projection for Dimension Reduction (UMAP) (McInnes et al., 2018) (box 2-b in Figure 1). This reduces the complex data while preserving its essential structure and improves the performance in the next steps. On this simplified representation we apply the HDBSCAN clustering algorithm to group

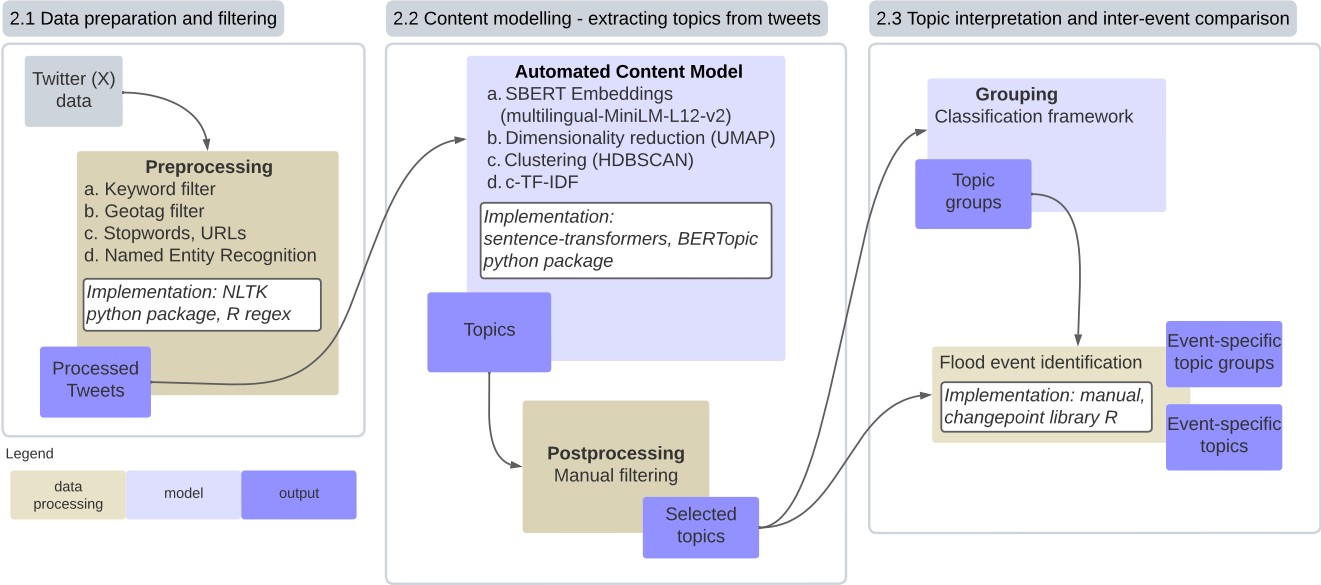

**Figure 1.** Flowchart of the topic modeling analytic framework developed in this study.

similar embeddings together, forming clusters that represent distinct topics within the data (box 2-c in Figure 1). (3) The last step (*Topic interpretation and inter-event comparison*) facilitates the identification of common themes and subjects discussed in the text. The clustered topics are refined through post-processing, where undetected noise and irrelevant information are further filtered out (box 3, Figure 1). This step ensures that the extracted topics are meaningful and relevant to the research objectives. For interpretation purposes, we apply a manual process to assign a meaningful category to each cluster (box 4, Figure 1). Here, we provide context and interpretation to the identified topics, aligning them with a state-of-the-art classification framework (Houston et al., 2015). Additional information on implementation and software is available in Supplementary Material Section 1.

## 2.1 Data Collection

The specifics of data collection can be found in de Bruijn et al. (2017, 2019). The following section describes the processing performed by de Bruijn et al. (2017, 2019) followed by an overview of the additional processing performed in this study, which is described in detail in Section 2.2. The full data was collected based on the former Twitter (X) API in eleven languages (de Bruijn et al., 2019). The data collection and processing involve three main types of input data. First, the authors of de Bruijn et al. (2017, 2019) used a database of known geo-locations, which contains over 4 million geographical locations including cities, towns, villages, and administrative divisions, along with alternative names and translations. Second, they collected tweets and associated metadata in real-time through the Twitter (X) streaming API using flood-related keywords in eleven languages, gathering 55.1 million tweets between July 2014 and July 2017. The keywords included terms like "flood," "flooding," and

"inundation" and their equivalents in other languages. Third, they utilized GIS shapefiles of global time zones and analyzed Wikipedia articles to obtain lists of the 1000 most commonly used words per language (excluding location names with populations over 100,000). The data processing involved matching tweet text to the gazetteer through toponym recognition, scoring candidate locations based on spatial indicators, grouping related tweets, and using a voting process for toponym resolution. The system processes tweets in 24-hour windows and maintains a toponym resolution table to enable real-time geoparsing of new incoming tweets. Relevance to flooding was further ensured by classification and pre-selection based on BERT.

Based on this data we additionally performed a combination of keyword and geolocation searches during the data pre-processing to obtain tweets related to flooding events in our study areas. We analyze a sample of textual Twitter (X) posts (n=43,287) collected from 2014 to 2022. Our sample includes all tweets posted during this time containing one or more of the three flood-related keywords (Hochwasser, Überflutung, Flut) written in German and geotagged within Germany. The table for all keywords in other languages is available at: https://www.nature.com/articles/s41597-019-0326-9/tables/2160 (de Bruijn et al., 2019).

## 2.2 Data Preparation and Filtering

Before passing the data to our modeling pipeline we performed several cleaning, filtering, and preprocessing steps. First, posts are eliminated based on 13 keywords that indicate non-flood related contexts. For example, any tweet containing variations of the word *"flood of skilled workers"* (*"fachkräfte-flut"*) is removed from the dataset. The keywords were identified in the exploratory data analysis when screening the texts. The second, we remove URLs and stop words from the remaining tweets based on a dictionary of German stop words. To avoid creating topics based on frequently mentioned locations or users while keeping sentence structure intact, we replace mentions of locations of users with general example. We replaced locations with the German word describing the NUTS3 region associated with the respective geotag. The geotags linked to each tweet available and extracted according to the method proposed by de Bruijn et al. (2017). The removal was performed by matching the identified words with the words within the tweet. If a user was tagged specifically with their username (@thisusername-wastagged), we replaced the username with the German word for user (Benutzer). Details about how often Twitter users post are elaborated in the Supplementary Material Section 2.2. We removed all other entities, such as names of people, places, organizations automatically after named entity recognition was performed. In this pre-processing step we tokenized the tweets and performed a part of speech tagging, where each chunk of a sentence is labelled according to its grammatical function. Those words labelled as entities were removed from the text. The resulting preprocessed Tweets are then passed to the automated content model.

## 2.3 Content Modelling

In the following we formulate and describe the methodological details of the transformer embedding, clustering steps, and the class-based Term Frequency Inverse Document Frequency (c-TF-IDF). BERTopic algorithm represents the fully automated core of our proposed framework. We interpret the automatically formulated topics in Figure 3, 4, and Sections 3.2 and 3.3. Results in Figure 3 and 4 are independent of the manual classification that follows in the results Section 3.4 and Figure 5.

## a. SBERT

We process the tweets with a pre-trained transformer model (SBERT, version:paraphrase-multilingual-MiniLM-L12-v2), which creates a 384 dimensional dense vector representation of the tweets (Reimers and Gurevych, 2019). SBERT is an extension of BERT (Devlin et al., 2019), which is optimized for classification or clustering semantically similar sentences. SBERT is suitable for our study, since we aim to cluster the embeddings to extract topics, which represent tweets with similar content. While SBERT is pre-trained on general-purpose datasets, we found its performance on our disaster-related corpus to be robust. For verification, we conducted an experiment where we compared a German model (GermanBERT) (Darji et al., 2023) and a model trained on Tweets specifically (TwHIN-BERT) (Zhang et al., 2023), where we found that the topics were less distinct and interpretable. Thus, we proceeded with the pre-trained SBERT model to maintain generalizability, aiming to demonstrate an approach which, in the future, can be adapted to different contexts and case studies.

## b. UMAP and c. Hierarchical Density based Clustering

As clustering performance has shown to reduce in high dimensional space (Allaoui et al., 2020), we reduce the embeddings to a 3-dimensional space using UMAP (McInnes et al., 2018). The reduced embeddings are categorized with Hierarchical density based clustering (HDBSCAN) (McInnes et al., 2017). More information on the hyperparameter tuning is described in the Supplementary Material Section 2.2. To evaluate the quality of our HDBSCAN clustering, we calculate the Density-Based Clustering Validation (DBCV) score (Moulavi et al., 2014). Based on the definition that clusters represent areas of higher density amongst regions of lower density, a relative validity measure is calculated by combining the shape and density properties of the cluster. The density is evaluated relative to the density in a cluster representing the background noise. We obtained a positive score of 0.24 within the DBCV range of -1 to 1, which validates the effectiveness of our clustering approach in identifying distinct topics within the tweet corpus.

While the thorough preprocessing significantly improved our textual data quality, some unstructured, non-actionable textual Twitter(X) posts will still show up in clustering. The chosen BERT embeddings are robust when confronted with word substitution attacks (Hauser et al., 2021). Further, we chose HDBSCAN, which combines hierarchical clustering to avoid ambiguity and density-based methods to account for the noise in the dataset. In the first clustering step, the denser areas are separated from the surrounding points to separate areas of interest from the background noise, that is, in our case, unstructured texts. Following the separation of unstructured text and clusters of similar content discussed often, a minimum spanning tree is constructed based on a weighted graph containing the embedded textual tweets as vertices and their weighted connection based on the mutual reachability distance. Based on this we construct a hierarchy of connected components, which is then used to cut the dataset into clusters within the hierarchical structure. These steps minimize ambiguity between the clusters by using a condensed clustering tree and defining the clusters by minimum cluster size (in this case 20). Our further analysis accounts for this limitation by focusing on aggregate trends rather than individual posts.

#### d. Class-based Term Frequency-Inverse Document Frequency

In the next step we aim to understand the meaning of each topic by representing a topic with 10 key words. The representative words may contain two consecutive words as one keyword. To achieve this we use a class-based Term Frequency-Inverse Document Frequency (c-TF-IDF) as proposed by Grootendorst (2022). All tweets from the same cluster are combined and treated as one document. With this representation the c-TF-IDF of a word $x$ in cluster $c$ ($W_{x,c}$) is calculated as described in Equation 1. The c-TF-IDF is calculated based on the frequency of a word in all classes, frequency of words ($tf_{x,c}$) within a cluster and mean number of words ($A$) within a class.

$$W_{x,c} = |tf_{x,c}| * log(1 + \frac{A}{f_x}) \tag{1}$$

**Postprocessing**

With this approach we obtain a large number of topics, that were passed to a post processing pipeline. Similar to the filtering steps in the preprocessing, we manually scan and exclude the topics based on whether the keywords indicate flood-related content. Additionally, topics with fewer than 50 instances over the whole time span are excluded in this analysis. To aid the inter-event comparison we adopt a functional framework for social media use from Houston et al. (2015). The authors proposed that social media can have 15 types of functions that are associated with the three phases of an event (pre-event, event, post-event). Pre-event the tweets can be used to spread preparedness information or provide warnings. Shortly before or once the event started users can signal and detect the disaster on social media. During the event, requesting help and sharing condition and location of flood affected individuals become more important. Documentation, consuming news coverage, receiving response information, volunteering, receiving health support as well as expressing emotions and sharing stories about the disaster happens during and after the event. Post-event tweets can start discussions on scientific and socio-political causes as well as connecting community members and coordinate the implementation of traditional crisis communication activities. We manually associate the topics obtained from our model with their respective function in the framework. We refer to the direct model results as *topics* and to the classified topics as *topic groups*. We manually classify all topics with 50 instances or more into the topic groups. To counteract confirmation bias, we assign the topic groups before we examine the temporal results.

### 2.4   Topic interpretation and inter-event comparison

To evaluate our model we follow a "zoom-in" approach to gain insights at varying levels of detail and context. Initially, we analyze the entire time series but divide it into periods of flooding and non-flooding as a baseline. We observe distinct topic patterns in Twitter (X) by comparing the topics and topic diversity for the two subgroups. Next, we narrow our focus to the weeks around five distinct flood events, comparing how individual topics evolve over time during these periods. With this approach, we evaluate which topics arise commonly and how they vary across different flood types by looking at specific topics over time. Lastly, we aggregate topics throughout the entire event duration to compare broader categories. This allows us to compare the general topics across different flood types.

## 2.5 Flood Events

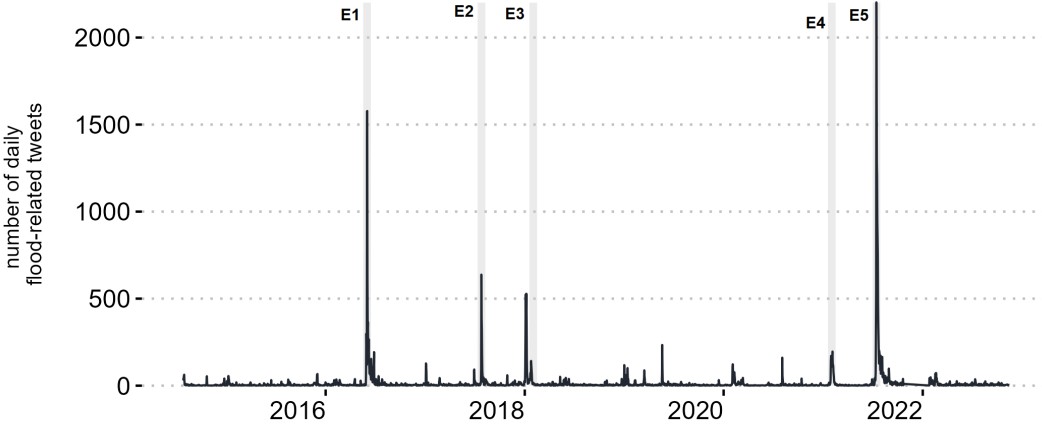

**Figure 2.** Daily number of tweets over the observed time period (black line). The gray lines labeled E1-E5 mark the occurrence of the selected flood events within the time series, which are further described in Table 1. The shaded areas show their time frames and highlight the specific peak time we consider for the selected flood events.

We analyze a sample of textual Twitter (X) posts (n=43,287) collected from 2014 to 2022. Our sample includes all tweets posted during this time containing one or more of the three flood-related keywords in German (*Hochwasser, Überflutung, Flut*). The table for all keywords in other languages is available at: https://www.nature.com/articles/s41597-019-0326-9/tables/2 (de Bruijn et al., 2019). Figure 2 shows the number of daily tweets we used for our analysis after the initial filtering steps. We selected five events between 2016 and 2022 (Table 1). Based on these events we will qualitatively evaluate our approach and results. The most discussed flood in our dataset (*E5*) occurred in July 2021 in Europe and Western Germany. This event was caused by the atmospheric low-pressure system *Berndt*, which brought heavy rainfall to two German federal states as well as adjacent countries (Luxemburg, Belgium, and the Netherlands) (Mohr et al., 2022). The flood caused 189 fatalities and losses of around 33 billion Euros in Germany, making it the most severe natural disaster in recent German history (MunichRe, 2022). In 2016 persistent atmospheric conditions triggered a large number of heavy convective rainfall events, resulting in local but extreme flash floods, particularly affecting the towns of Simbach in Bavaria and Braunsbach, in Baden-Württemberg (*E1*). These events caused 54 fatalities in Simbach and substantial economic damage in both towns (Laudan et al., 2017; Hübl and Rimböck, 2018; Bronstert et al., 2018). The flood events were associated with a return period above 100 years and the discharge of the Simbach Creek was further increased by dam and dyke failures (Hübl and Rimböck, 2018). E1 and E5 represent flash floods with high impact in terms of fatalities as well as economic damage that occurred in our observation period. During both events the peak daily tweet frequency exceeded 1500 $\left[\frac{tweets}{day}\right]$

On the 25th of July 2017, the area between Goettingen and Brunsvik in Lower Saxony was affected by a flood (*E2*) caused by three-day continuous rain due to the low-pressure system "Alfred". In the Nette and Oker rivers, two gauges reported return

**Table 1.** Features of the five flood events selected for comparison in this study

| Id | Gauge, River | Date of peak discharge | Return period [years] | Flood type | Reference |
|----|--------------|------------------------|-----------------------|------------|-----------|
| E1 | Simbach, Simbach | 1st of June 2016 | >100 | severe flash flood | Hübl and Rimböck (2018) |
| E2 | Ohrum, Oker | 27th of July 2017 | 50 | medium impact riverine flood | Anhalt et al. (2017) |
| E3 | Maxau, Rhine | 25th of Jan 2018 | 10 | frequent riverine flood | Helmke et al. (2018) |
| E4 | Schotten I, Nidda | 4th of Feb 2021 | 25 | frequent riverine flood | Löns (2021) |
| E5 | Ahrweiler, Ahr | 15th July 2021 | >1000 | severe flash flood | Mohr et al. (2022) |

periods of 100 years and on the Innerste River, two gauges reported even higher return periods (Anhalt et al., 2017). No fatalities occurred, however, 12 individuals were displaced by the flood (Brakenridge) and reported damages were in millions of euros (Anhalt et al., 2017). In our topic analysis, E2 is evaluated separately representing a medium impact event.

In January 2018, torrential rains and storms combined with snow melt resulted in high water levels in many German regions, with moderate floods (maximum return period of 10 years in Maxau, Rhine) (*E3*) (Helmke et al., 2018). In the last week of January and the beginning of February 2021 continuous rain along with a thaw period led to increasing discharge in Hesse (*E4*) (Löns, 2021). In February 2018 the municipality of Buendingen was affected increasingly by the flood and 70 people were evacuated from the old town (Löns, 2021). The discharge in the respective river Nidder exceeded a return period of 100 years, most of the other rivers in the region experienced maximum discharge of return periods between 2 and 50 years. E3 and E4 represent events that are expected to occur more frequently with lower impact in terms of monetary damage and fatalities.

To evaluate the topic model we compare the topics group frequency for the high impact flash floods E1 and E5, moderate flooding and low-impact riverine flooding and the temporal development of topics over time for E3 and E4.

Flood severity was classified based on on official warning levels by LUBW (2024) using return periods of water levels. This classification corresponds to the official warning levels in Germany.

# 3 Results

## 3.1 Full dataset results

Our first key finding from the Tweet analysis shows that approximately 78% of the analyzed tweets contain valuable information for disaster management. While this is a promising result to dig further in the following topic extraction and analysis phase, it also shows that a non-negligible portion of our Tweet posts in our dataset is classified as noise or irrelevant information

despite the thorough selection of tweets according to flood related keywords. 10,183 tweets are identified as noise by the algo-rithm. 7233 tweets belonging to 34 topics are manually removed (postprocessing in Figure 1). Those tweets are not considered for our further analysis due of their lack of meaningful content with respect to our research objective. Supplementary Material Figure S2 shows the temporal development of monthly tweets that were not assigned to a relevant topic. Here, we find that the progression of the noise in the data follows the path of the daily time series (see Figure 2). This leads to the conclusion that

noise is proportionally equally distributed during the selected events and the baseline.

      Over the whole time period of our analysis, we found 500 distinct topics in flood related tweets. To refer to topics in this section we use the numerical topic ID followed by the most accessible keyword or element from the representative tweets reported in Supplementary Material Table S1 and S2 (for example topic "T-0-information", topic "T-1-weather extremes". The topic ID starts from 0 and is inversely correlated to the number of tweets assigned to the topic across the entire temporal span.

Consequently, topic "T-0" shows the highest tweet count, while topic "T-489" achieves the lowest incidence over the course of five years. Specific topics are analyzed in Figures 3 and 4.

### 3.2   Aggregated topic analysis

    As a first step to analyze the content of tweets, we focus on the topics that were most frequently observed in a single day (Figure 3). The numbers on the bars represent the topic labels, with labels increasing in value from the most to the least frequent across

the entire time period. Each bar shows an event period or the event-free period. Therefore, if the same topic numbers appear in different bars, the content of tweets during these events is similar. Figure 3 shows that the most frequent topics over the whole time period (topic "T-0 water authorities" and" T-1 weather extremes") are also represented in the maximum daily occurrence for E3 and E4, which form the low-impact event group. topic "T-0 water authorities" and "T-1-weather extremes" primarily contain tweets that describe reports of water levels (representative tweets: *"pegel bundesland aktuelle hochwasser*

*info liegt vor mehr unter",* *"water gauge federal state current flood more info available below"*). These Topics are mostly linked to generic posts on water levels as posted by *(@) flood portal / hochwasserportal_de* and then shared among users. Event-free times are marked by a consistent, small number of tweets related to topics 0-information, 1-weather extremes, 2-warning, and so on. This pattern leads to a high overall sum (as shown in Supplementary Material Figure S1), but with only a few daily occurrences. Topics that received the highest daily attention on Twitter (X) for E1 and E5 which represent

high impact flash floods are related to reports of fatalities and missing people (topic "T-110 deaths", representative tweet: "three dead in flood in district", "T48 destruction", representative tweet: "four dead in flood disaster in federal state", and "T-7 disaster management",representative tweet: "civil protection rehearsed the emergency months ago and failed flood") as well as discussing political implications (topic "T-24 chancellor",representative tweet: "new contribution after flood Merkel in city Tagesschau", and "T-122 euro",representative tweet: "Soeder announces euro immediate aid for flood victims federal state pays

the affected"). A full list of representative tweets and keywords of the topics mentioned in this paragraph is appended in the Supplementary Tables S1 and S2. In contrast to the topics observed for E3 and E4, these are event-specific topics that are most likely shared due to personal concern and shock. During E2 (medium impact flooding) the discussion on Twitter (X) is focusing on event-specific topics that describe impacts (topics"T-152" dam guards,representative tweet:"flood in country the night was

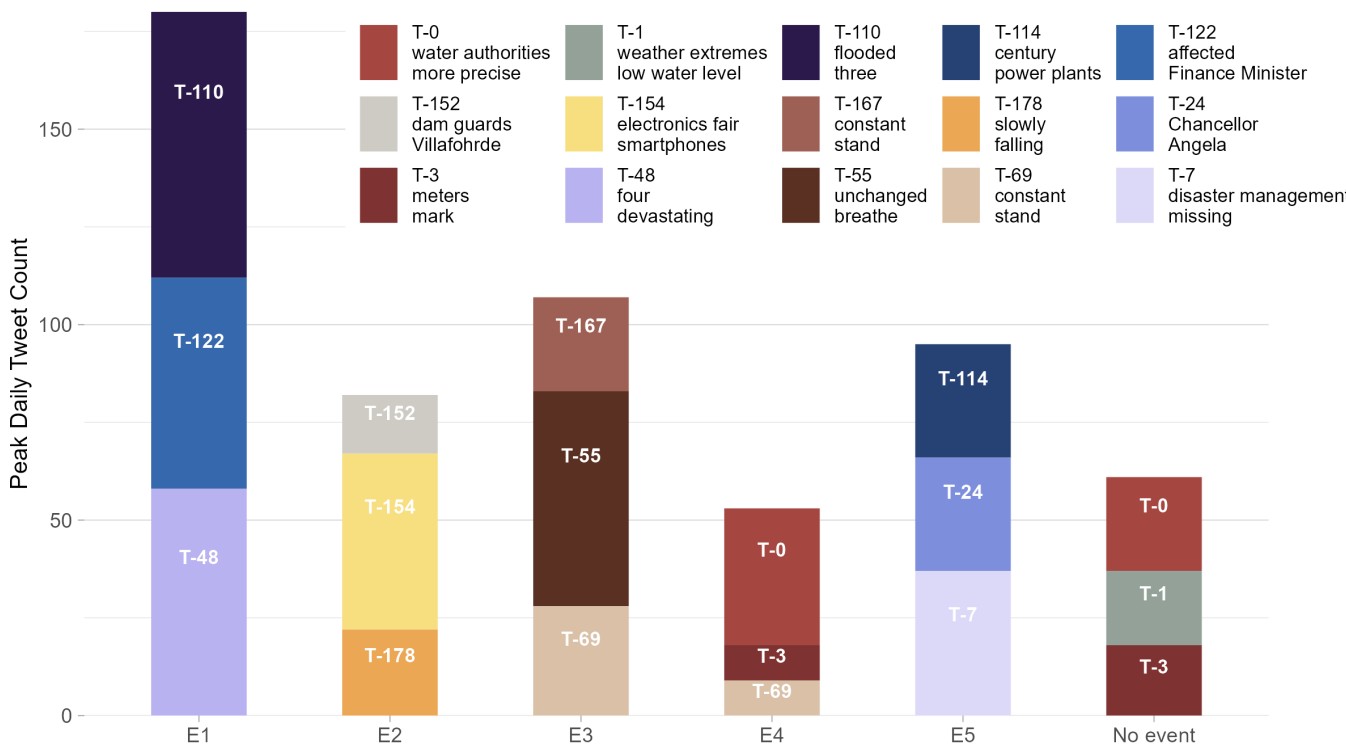

**Figure 3.** Stacked bar chart for topics that occurred most frequently in a day during the baseline period (no event) and the different flood events (E1-E5). E4 and times without events show textual Twitter (X) posts with the topic administrative updates (T-0: water authorities), weather conditions (T-1: weather extremes), and infrastructure status (T-3: meter marks), E5 also includes a topic related to infrastructure status (T-114: power plants). During E1 disaster impacts (T-48: devastating damages, T-110: flooded) are discussed. E5 includes a topic on emergency response (T-7: disaster relief) while the retreat of water levels (T-55: unchanged) is mentioned frequently during E3. The topics presented in this figure are the results from the HDBSCAN Clustering and c-TF-IDF analysis

calm flood in federal state is somewhat relaxing", "T-154 electronics fair", representative tweet:"flood district declares disaster alarm", and "T-178 falling", representative tweet: "flood in federal state water levels are dropping only slowly"). The wording of topics related to E2 is not predominantly generic like for the low-impact flood events, but still remains pragmatic, analytical, and descriptive compared to E1 and E5.

Overall, we observe a shift in tweet topics from spreading general information about water levels to discussing more complex and impact-focused topics during events. To gain a better understanding of this dynamic we further undertake a temporal analysis to understand the content shared over time during the different phases of a disaster.

Additionally, aggregated topic patterns during floods are characterized by the number of different topics that occur in a 40-day time window enveloping the flood peak (see Table 2). Events that are predominantly flash floods with a higher impact

**Table 2.** Number of different topics that occur in a 40-day time window enveloping the flood peak.

| Id | Number of topics |
|----|------------------|
| E1 | 132 |
| E2 | 109 |
| E3 | 73 |
| E4 | 89 |
| E5 | 128 |

result in a wide range of topics (E1: 132, E5: 128). Lower-impact riverine flood events resulted in fewer different topics discussed on Twitter (X) (E2:109, E3:73, E4:89). Moreover, the distribution of topic appearances within high impact floods is more heterogeneous. Supplementary Material Figure S3 shows the distribution of the count of all topics for E4 and E5, both of which occurred within the same year and region. To mitigate the potential noise introduced by topic diversity, we used the HDBSCAN clustering algorithm, which contains a hierarchical topic extraction step. Primary topics, i.e., topics which occur consistently, are given higher weights in the methodology, based on their mutual reachability distance. Topics with a lower relevance, while contributing to the overall understanding of the event, were given lower weights when condensing the clusters. This approach allows us to maintain focus on critical information while still capturing the broader context of the event. For E4 there are three distinct peaks in the chart indicating a focus on few topics within the 89 total topics. For E5 we see many peaks, indicating frequent occurrence of many different of the 128 topics. This shows that the tweet content of high impact events is more diverse and complex. These findings suggest that topic diversity might be used as an indicator to rapidly predict flood impact. We observe that the presence of greater topic diversity in tweets may be indicative of a potential for high impact events.

## 3.3 Temporal topic analysis

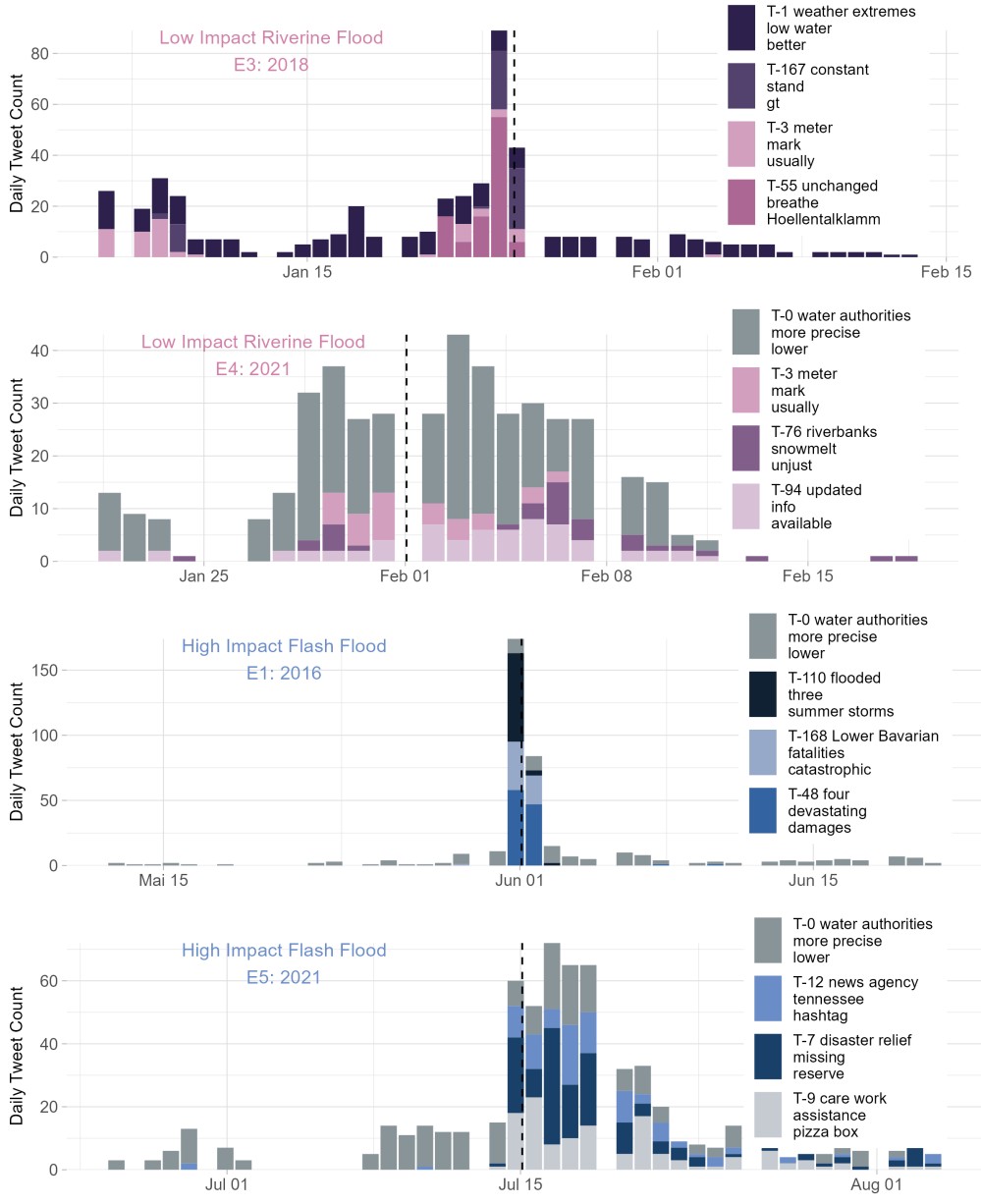

**Figure 4.** Progression of tweet count per topic for low-impact riverine floods (E3, E4) and high impact flash floods (E1, E5). The dotted line in each subplot represents the time of observed peak discharge. The legend indicates different topics of textual Twitter (x) posts, including: weather conditions (T-1: weather extremes), administrative updates (T-0: water authorities), infrastructure status (T-3: meter marks), general information updates (T-94: updated info), disaster impacts (T-48: devastating damages, T-110: flooding), emergency response (T-7: disaster relief). The topics are the result of the HDBSCAN clustering and c-TF-IDF analysis.

The temporal evolution of tweet activity and content on Twitter (X) varies significantly depending on the type and impact of the flooding event. Figure 4 shows the event time window for the events categorized as low-impact riverine flooding and events categorized as predominantly flash floods with high impact. Twitter (X) users engage differently on Twitter (X) during flash floods compared to riverine flooding. When it comes to flash floods with a high impact, a surge in tweet activity is observed shortly after the peak discharge occurs. In contrast, for riverine flooding, we note a gradual increase in tweet activity that begins days before the flood event. Additionally, we find a sharp decline in the discussion of valuable topics following the peak discharge. This indicates that social media platforms may be exploited for immediate response and coordination, with limited utility for preparedness or long-term recovery activities.

The content of these flood-related tweets varies for the different flood types. Supplementary Material Table 2 offers detailed descriptions of all topics including representative tweets. For Events E3 and E4 (Figure 4), the progression of topics spans the entire duration of the event, with a focus on aspects like water depths and natural processes (E3: topic "T-55 unchanged", E4: topic "T-76 snow melt"). Especially topics "T-3 meter" and "T-94 updated" point towards more generic Twitter (X) content. The consistent presence of these topics throughout the event timeline suggests that, during low-impact flooding, people tend to be more proactive and prepared. They actively share information before, during, and after the flood, in contrast to impulsive tweeting when they are directly affected by the event. Here, topic "T-55 unchanged", referring to the stagnation and decline of water levels, is an exception since during the 2018 flood there was a peak of tweets indicating that a previous warning or alarm had been lifted.

During high impact flood events, people start discussing topics like reporting fatalities, offering help (topic "T-9 care work"), disaster management (topics "T-7 disaster management" and "T-168 fatalities"), and sharing traditional media like newspaper articles (topic "T-12 news agency"). This shift is clear in the lower plots in Figure 4, which highlight the four most common topics for E1 and E5. The wording in the topics for high impact flood events is more impact-focused ("damages", "fatalities") and urgent ("missing", "reserve") or even catastrophic ("devastating", "catastrophic").

The progression of topic "T-9 care work" in Figure 4 (extract from representative tweet: *"concerns city targeted offers of help are collected under URL...", "betrifft kreis stadt gezielte hilfsangebote werden unter URL gesammelt..."*) shows that Twitter (X) is used to coordinate response activities. The topic emerges predominantly after the peak discharge of E5. This self-organized disaster response on Twitter (X) can potentially be channeled and used as an information source for organizationally coordinated response activities.

For E5, we initially see fewer than 15 daily posts related to topic "T-0 water authorities", which is similar to the number of textual Twitter (X) posts during non-flooding times. The discussion on flood-related topics starts suddenly on the day of the flood event. This timing matches previous evaluations of how well the emergency management and warning system worked, as discussed by Thieken et al. (2022). This finding shows that the topics retrieved with our approach reveal real-world flood aspects problems with early warning reported in this case.

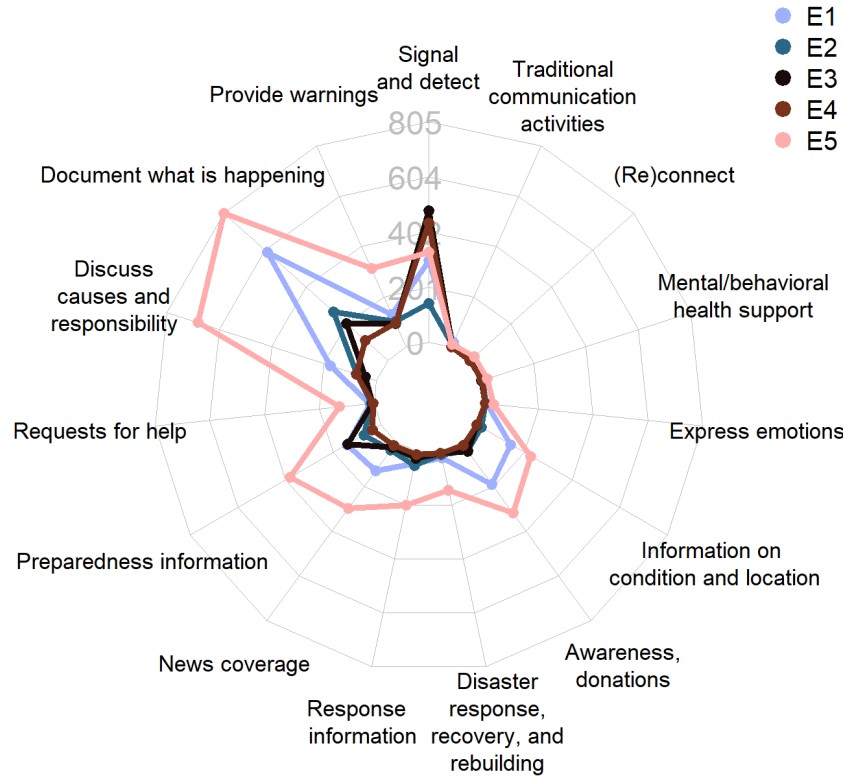

**Figure 5.** Representation of topics categorized according to the 15 functions proposed in the functional framework for social media usage types during disasters by (Houston et al., 2015). The results presented here are a manual aggregation of the Topics presented in previous figures.

### 3.4 Event comparison within a state-of-the-art functional framework for topic classification

By grouping the topics within an established framework we qualitatively validate our results and put them into context with findings from other studies. In this step, we consider topics with more than 50 instances over the whole time series. Houston et al. (2015) outlined fifteen distinct functions of social media, as detailed in the Materials and Methods section. We categorize the topics we found in our previous analysis according to these different functions and display the resulting functional

distribution in Figure 6. Within our dataset, we did not find evidence of Twitter (X) being utilized for four of these desig-
nated functions (implementing traditional communication activities, (re)connecting community members, health support, or
generally expressing emotions).

A limitation to the applicability of our model to different platforms and circumstances is the need for manual filtering and
the associated uncertainties. The manual steps limit the transferability and may introduce a bias due to the individual variability
of keyword selection. This limitation can be addressed by improved or combined embedding models (Laskar et al., 2020) or
an embedding-based pre-selection.

E3 and E4 have most tweets related to topics with the function of signaling and detecting disasters. During E2, which we
view as a moderate impact event, most of the tweets were assigned to topics, which had the function of documenting what is
happening in the disaster, and a slight indication of preparedness information being shared on social media. E1 shows a similar
pattern with a higher magnitude and an increasing interest in documenting the flood. During E1 Twitter (X) users also started
to discuss the socio-political impacts and responsibilities and shared links to traditional news outlets.

During E1 we also observe an emergence of topics that indicate awareness and financial support. With the increasing impact
of the flood event, we can see the progression of this trend for topic groups. For E5 we see that the focus lies on documenting
the disaster and discussing socio-political responsibilities and a further increase of interest in the topic groups that emerged for
E1.

These findings underscore the substantial shifts in topics and topic groups associated with events of varying impact and
magnitude.

## 4 Conclusions

In this study, we develop a transferable natural language processing analytic approach for automatic extraction of content from
flood-related social media posts collected over a multi-annual time period. Our approach is based on openly available software,
data and pre-trained models making it accessible to researchers and users.

Despite the general value and applicability of our proposed approach, along with our key findings, our analysis is associated
with uncertainties and can be further improved. The pre-trained transformer model by Reimers and Gurevych (2019) and the
quality of the clustering of the embeddings extracted from the transformer encoder provide the basis for the quality of topics
that are extracted. Therefore, this methodology may be improved with the development of large-scale language models that
focus on clustered encoding. The identification of distinct topics within tweets is susceptible to the possibility of topic overlap
and separation. By using the HDBSCAN clustering approach we are setting the level of topic separation through the minimum
cluster size, which is a rigid threshold technique. Future model refinement should emphasize strategies to allow more flexibility
while ensuring topic separation and improving the clarity and robustness of topic-based interpretations. A similar issue arises
with topic representation. We represented the topics with ten keywords alongside a representative tweet. Defining the meaning
of this combination of keywords is a subjective task that leaves room for different interpretations, leading to uncertainty. Here,
the framework is also limited in terms of expert-based manual steps, for example for the topic exclusion after modeling and

the classification in an existing framework. We acknowledge that this process would benefit from multiple people labeling topics independently, however, the team of authors continuously discussed the topic assignment and exclusion in the process. The embeddings retrieved from the sentence transformer model vary slightly for each run. More in detail, this may cause slightly different results on tweets at cluster borders when re-running the framework. However, this does not notably affect the topic size and representation in this study because of the robustness of the HDBSCAN algorithm. Furthermore, social media platforms like Twitter (X) represent only a subset of society. Consequently, insights drawn from textual Twitter (X) data may not fully capture the different experiences, perspectives, and actions present within the broader population. This limitation can be addressed by evaluating the topics discussed on alternative social media platforms other than Twitter (X), which also ensures robustness towards the fluctuations of access to data from individual providers. In de Bruijn et al. (2019), Figure 3 the textual Twitter (X) data is successfully validated against NatCatSERVICE data provided by Munich Re, further ensuring robustness. Additionally, we recommend conducting an analysis taking the geolocation of tweets into account and evaluating to which extent the identified social media topics can improve flood models and simulations predicting, for example, flood impact measures such as expected damage by including tweet topic distribution alongside tweet counts, similar to e.g. Re et al. (2022). Non-textual information contained in social media posts could be included to reduce uncertainties associated with unstructured grammar or spelling mistakes, for example by comparing the image content and text content of a post

We show that the proposed methodology and extracted content allow discovery of citizens' behaviour and perception of floods before, during, and after different disaster types. We successfully validated our model results qualitatively, based on previous knowledge about past events. Approximately 78% of tweets contain potentially valuable information for flood risk management, which indicates an opportunity to encourage social media users to share flood related content online. Our results confirm that there are distinct topic patterns in the Twitter (X) time series. These patterns are associated with a shift of tweets focused on sharing generic information and warning topics towards more diverse topics including coordination of response activities and more complex discussions surrounding the event. This shift is confirmed when looking at different event types. From low to high impact events we see a progression in the number of topics as well as a progression in the content from signaling the disaster to discussing causes and responsibilities as well as documentation activities. This shows the importance of social media in the response process. However, we see that the potential for coordination of immediate response activities is the most promising risk management intervention, as long-term activities such as rebuilding or remembering floods are not visible in our dataset. During low-impact riverine floods citizens are responding more routinely and information on water levels are effectively shared on Twitter during E3 and E4. Therefore, we recommend including spreading information on social media platforms for early warning and risk management strategies. We show that risk management interventions on social media can be supported through institutional posts. This appears to be particularly helpful for immediate response coordination, particularly during the time shortly after a high impact flood where citizen engagement is high and organization-focused. Moreover, spreading water level information as soon as it becomes available before the event of flash floods is recommended to shift awareness dynamics towards those exposed during riverine floods with lower impact. We find that we can partly reproduce functions that were attributed to social media use during disasters in a theoretical framework, for flood events in Germany. However, not all assumptions Houston et al. (2015) made about social media usage during disasters can be shown in

our dataset. This might be because some functions of social media are platform-dependent and therefore outside of our horizon of observation. The proposed methodology, however, is not limited to be applied to tweets but can be applied to any text-based social media platform with an accessible API, which is extremely useful in the context of quickly evolving and changing online

platforms.

*Code and data availability.* The scripts used for the analysis are available at: https://github.com/SWN-group-at-TU-Berlin/SocialMediaNLP_ FloodTopics. Tweet location identification is based on an algorithm developed by de Bruijn et al. (2017). The dataset of flood related Twitter posts from de Bruijn et al. (2019). The Data is available at https://www.globalfloodmonitor.org/ upon request.

*Author contributions.* **Nadja Veigel**: Conceptualization, Methodology, Software, Validation, Formal analysis, Visualization **Heidi Kreibich**:
Conceptualization, Writing - Review & Editing, Supervision **Jens de Bruijn**: Data Curation, Methodology, Conceptualization, Writing - Review & Editing **Jeroen C.J.H. Aerts**: Conceptualization, Writing - Review & Editing, Supervision **Andrea Cominola**: Conceptualization, Methodology, Writing - Review & Editing, Supervision

*Competing interests.* One of the authors is a member of the editorial board of NHESS.

*Disclaimer.* During the preparation of this work the author used ChatGPT to improve language. After using this tool/service, the authors
reviewed and edited the content as needed and take full responsibility for the content of the publication.

*Acknowledgements.* The authors would like to thank the Helmholtz Einstein International Berlin Research School in Data Science (HEIB-RiDS) for supporting this project.

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
