# Peer review of "Content Analysis of Multi-Annual Time Series of Flood-Related Twitter (X) Data"

_EGUsphere, 2024_

## Referee Comment (RC2)

Referee report on the manuscript
**"Content Analysis of Multi-Annual Time Series of Flood-Related Twitter (X) Data"**
submitted to the journal
**Natural Hazards and Earth System Sciences**

The authors have undertaken a relevant study by leveraging social media data (X, formerly Twitter), to detect topics related to hazards using advanced techniques like SBERT and clustering. The use of pre-trained language models and unsupervised clustering methods demonstrates a sophisticated approach to analyzing the vast and complex data generated on social platforms during disasters. This research contributes valuable insights into how social media can be harnessed for disaster management and impact assessment, and the methodology employed has the potential to improve disaster response strategies. Additionally, the paper is well-structured, enhancing readability and scientific clarity. The figures and tables are clear and effectively represent the results, aiding the overall understanding of the study.

Following are a few of the concerns that require clarification.

1.  Data collection from X (formerly Twitter) has become increasingly difficult and costly. In earlier versions of the Twitter API, researchers could easily gather data using queries and relevant keywords without limitations, which contrasts sharply with the current restrictions on the X developer platform. Including a detailed explanation of your data collection strategy, ideally in Section 2, would be highly beneficial. I understand that some data and data collection details are mentioned in section 2.4. However, including how the data was gathered will be useful.  It could serve as valuable guidance for researchers working in the social media field, helping them navigate these new challenges.

2.  In Table 1, the events E1, E2, E3, E4, and E5 are well-documented. However, I am curious about how the severity or impact of the floods was measured. Was this evaluation based on factors such as the number of casualties, the extent of the affected areas, or other key indicators? Clarifying this would provide a clearer understanding of the flood classification.

3.  While the observation that greater topic diversity in tweets may indicate high-impact events is intriguing (mentioned under section 3.1), I have some concerns regarding the potential for ambiguity in the analysis of specific

disaster impact factors, such as casualties or damages. The detection of a higher number of topics may introduce noise and reduce clarity when focusing on key indicators of event severity. In particular, diverse topics may dilute attention on actionable information and make it more difficult to derive clear insights related to specific impact metrics. I suggest that the authors address how they mitigate this potential ambiguity when assessing diverse topics. Clarifying whether any topic weighting or hierarchies are applied could strengthen the argument and ensure that topic diversity does not introduce noise.

4. While the observation that people tend to be more proactive during low-impact flooding is an interesting insight (line 250), I have some concerns about the challenges involved in processing social media data. During disaster events, social media posts often contain typos, grammatical issues, unstructured sentences, and sometimes even lack critical information such as location details. Additionally, many users share images or videos without accompanying text, further complicating the extraction of useful insights. Given these limitations, there is a high chance of missing relevant data if basic pre-processing techniques are used. It would be valuable if the authors could clarify the specific pre-processing methods they applied to address these issues. For example, how were posts with poor grammatical structure handled? Were posts with missing location data discarded or enriched through external methods? Moreover, did the analysis account for non-textual posts, such as images or videos, that might carry significant information? Addressing these points would enhance the robustness of the findings and provide a clearer understanding of how diverse social media data was managed during different phases of the disaster event.

5. The use of SBERT in this study is impressive. However, pre-trained models like SBERT are typically trained on general-purpose datasets. Given that social media data often contains domain-specific terms and jargon, particularly in the context of disaster management, SBERT may not be fully equipped to accurately represent these specialized terms. I recommend that the authors clarify whether they fine-tuned SBERT on disaster-related data or other domain-specific corpora to enhance its performance for this particular application.

6. While the use of SBERT and HDBSCAN for topic detection is appropriate, the paper does not provide any evaluation metrics to assess the performance of the clustering model. Including results from common clustering evaluation metrics, such as Silhouette Score, or others, would give a clearer indication of the quality of the identified clusters. I recommend that the authors include these metrics to validate the effectiveness of their clustering approach. I believe that some details might be available in supplementary materials as mentioned in line 125, but the links to these materials cannot be found in the document.

7. One of the main limitations of using social media data is its reliability and credibility, particularly in disaster management scenarios where the stakes are high. The potential for false information or rumors can undermine the effectiveness of response efforts. Therefore, it would be beneficial if the results of the model were tested against specific disaster events and compared with official records for validation. Such an approach would enhance the credibility of the findings and ensure that the insights derived from social media are both actionable and trustworthy.

Minor typos

1. Line number 60 - Reference is not given within brackets; Konya and Nematzadeh (2024).
2. Line number 98 - I assume the usage of the word "The" came in twice was a mistake.
3. While including in-text citations with multiple references (such as in line 15), the order of reference can be year-wise. This improves readability.

---

## Author Response (AR1)

Reply to the comment about the paper

**Content Analysis of Multi-Annual Time Series of Flood-Related Twitter (X) Data**

**Open Review Process, NHESS**

In preparing this response, we marked the original reviewers' comments in black while the authors' replies are in red. Line numbers refer to the original manuscript.

**Referee #1:**

R1.0: This review is concerned with the article titled "Content Analysis of Multi-Annual Time Series of Flood-Related Twitter (X) Data". It is divided into three categories, namely, general comments, specific comments and technical comments.

**General comments:** The title of the article "Content Analysis of Multi-Annual Time Series of Flood-Related Twitter (X) Data" clearly reflects the contents of the paper, and the abstract provides a concise, complete, and unambiguous summary of the work done and the results obtained. Both these sections are pertinent and easy to understand. The manuscript is well-written and well-structured, delivering the idea, methodology, and results clearly and concisely. The figures are descriptive and of high quality, and the tables are informative. It is well-referenced with proper credit attributed to previous and/or related works, and the authors indicate each of their contributions and competing interests. Crediting the use of AI tools such as ChatGPT is fantastic, we are conducting research in the age of the AI revolution. The paper presents a comprehensive and innovative approach to using social media data from Twitter (X) to understand human behaviour and perceptions during several types of flooding events in Germany. The study develops an approach using advanced natural language processing (NLP) techniques, leveraging pre-existing and accessible tools, including transformer-based models like SBERT and clustering algorithms such as HDBSCAN, to automatically extract flood-related topics from large social media datasets. Several steps to clean and filter the data have been presented. This allows for a nuanced analysis of public response to various flood events. The paper's relevance is clear, given the increasing reliance on real-time social media data for disaster risk management and the potential to enhance flood preparedness and response strategies. Thus, this manuscript has good scientific significance, scientific quality, and presentation quality.

We thank this Reviewer for their constructive evaluation of our work. The suggested amendments have been considered carefully and are addressed individually below.

Following are a few of the concerns that require clarification.

R1.1:

**Specific Comments:**

1. **Clarification on Data Filtering**: The process for removing irrelevant tweets is well-explained. However, more detail on the limitations of this filtering process could be helpful.
2. **Interpretation of Topic Groups**: The clustering approach is well-explained. However, further discussion on the specific implications of the topics identified (such as "disaster management" or "fatalities") could be more elaborated.
3. **Comparisons with Traditional Data Sources**: The paper highlights Twitter (X) data as an alternative to traditional flood impact assessments. What would be the difference between the results from social media and conventional data sources?

**Clarification on Data Filtering:**

This topic was also raised by another reviewer so we decided to add a new subsection in Section 2 titled "2.1 Data Collection", which is placed before the already existing section on "Data Preparation and Filtering". The new section will include the following information to provide clarity on the data source and therefore also on the necessary filtering and associated uncertainties:

"The specifics of data collection can be found in de Bruijn et al. (2017) and de Bruijn et al. (2019). The following section describes the processing performed by de Bruijn et al. (2017) and de Bruijn et al. (2019) followed by an overview of the additional processing performed in this study, which is described in detail in Section 2.2. The full data was collected based on the former Twitter (X) API in eleven languages (Bruijn et al. 2019). The data collection and processing involve three main types of input data. First, the authors of de Bruijn et al. (2017, 2019) used a database of known geo-locations, which contains over 4 million geographical locations including cities, towns, villages, and administrative divisions, along with alternative names and translations. Second, they collected tweets and associated metadata in real-time through the Twitter (X) streaming API using flood-related keywords in eleven languages, gathering 55.1 million tweets between July 2014 and July 2017. The keywords included terms like "flood," "flooding," and "inundation" and their equivalents in other languages. Third, they utilized GIS shapefiles of global time zones and analyzed Wikipedia articles to obtain lists of the 1000 most commonly used words per language (excluding location names with populations over 100,000). The data processing involved matching tweet text to the gazetteer through toponym recognition, scoring candidate locations based on spatial indicators, grouping related tweets, and using a voting process for toponym resolution. The system processes tweets in 24-hour windows and maintains a toponym resolution table to enable real-time geoparsing of new incoming tweets. Relevance to flooding was further ensured by classification and pre-selection based on BERT.

Based on this data we additionally performed a combination of keyword and geolocation searches during the data pre-processing to obtain tweets related to flooding events in our study areas. We analyze a sample of Twitter (X) posts (n=43,287) collected from 2014 to 2022. Our sample includes all tweets posted during this time containing one or more of the three flood-related keywords (Hochwasser, Überflutung, Flut) written in German and geotagged within Germany. The table for all keywords in other languages is available at: https://www.nature.com/articles/s41597-019-0326-9/tables/2160 (de Bruijn et al., 2019)."

Based on your comment we will add the following sentence on the limitations introduced by the initial filtering steps in section 4 after Line 299:

"A limitation to the applicability of our model to different platforms and circumstances is the need for manual filtering and the associated uncertainties. The manual steps limit the transferability and may introduce a bias due to the individual variability of keyword selection. This limitation can be addressed by improved or combined embedding models (Laskar, 2020) or an embedding-based pre-selection"

Laskar, M. T. R., Huang, J. X., & Hoque, E. (2020, May). Contextualized Embeddings based Transformer Encoder for Sentence Similarity Modeling in Answer Selection Task. In N. Calzolari, F. Béchet, P. Blache, K. Choukri, C. Cieri, T. Declerck, … S. Piperidis (Eds.), Proceedings of the Twelfth Language Resources and Evaluation Conference (pp. 5505–5514). Retrieved from https://aclanthology.org/2020.lrec-1.676

**Interpretation of Topic Groups:**

We understand that the reviewer is referring to the lack of clarity and depth in topic interpretation.

We would like to point the reviewer to Table S2 and Table S1 in the Supporting Information. The two tables contain topic descriptions for all topics represented in Figure 1 and Figure 2 in the main manuscript. To make the contained information more accessible we will translate the two tables to English in the revised manuscript, as the top words and representative text are currently in German. The additional information in the Supplement contains a representative tweet as well as 10 keywords associated with the topic. We chose to represent the topics in the main text by only one keyword to improve readability and conciseness. While this probably leads to an oversimplification of topic representation, we tried to counteract this with a few instances of stating representative topic tweets. In the revised manuscript we will improve the topic interpretation by (i) adding the representative tweet in the text more often, (ii) extending the description of the topics when mentioned for the first time, and (iii) referring to the supplementary tables more frequently in the text where helpful.

**Comparison with Traditional Data Sources:**

To discuss the reviewer's question, we will add the following sentence in the Introduction after Line 22:

"Social media captures immediate personal experiences and emotional impacts that might be overlooked in conventional assessments, but lacks the standardized methodology and detailed technical measurements found in traditional sources. Therefore, analyses of social media data should not be seen, but as complementary analyses that enhance traditional flood impact assessments by providing rapid situational awareness and capturing the social dimensions of flood impacts that might otherwise go undocumented."

R1.3:

**Technical Comments:**

1. **Grammar and Style**:
   o Line 98: "The The Second" should be "The second."

2. **Figure Labels and Descriptions**: The figures provide valuable visual insights, but some (esp. Fig 3 & 4) would benefit from clearer labels or captions, particularly where technical details like clustering results or topic distributions are involved.
3. **In-text Citations Formatting**: Ensure that citations within the text follow a consistent format. There are some minor inconsistencies in how sources are referenced throughout the manuscript.

We Technical Comments:

1. Grammar and Style:
   We will correct the duplicate "The" on line 98 and conduct another thorough proofreading of the manuscript.
2. Figure Improvements:
   We will enhance Figures 3 and 4 by:

   Including legend explanations in the caption and expanding captions to better explain technical elements (e.g., whether the represented results are an output of the clustering or other phases in our methodology).

3. Citation Formatting:
   We will review all citations to ensure consistent formatting throughout the manuscript according to the journal's guidelines.

Reply to the comment about the paper

**Content Analysis of Multi-Annual Time Series of Flood-Related Twitter (X) Data**

**Open Review Process, NHESS**

In preparing this response, we marked the original reviewers' comments in black while the authors' replies are in red. Line numbers refer to the original manuscript.

**Referee #2:**

R2.0: The authors have undertaken a relevant study by leveraging social media data (X, formerly Twitter), to detect topics related to hazards using advanced techniques like SBERT and clustering. The use of pre-trained language models and unsupervised clustering methods demonstrates a sophisticated approach to analyzing the vast and complex data generated on social platforms during disasters. This research contributes valuable insights into how social media can be harnessed for disaster management and impact assessment, and the methodology employed has the potential to improve disaster response strategies. Additionally, the paper is well-structured, enhancing readability and scientific clarity. The figures and tables are clear and effectively represent the results, aiding the overall understanding of the study.

We appreciate your thorough and constructive feedback on our manuscript "Content Analysis of Multi-Annual Time Series of Flood-Related Twitter (X) Data". Your comments have been helpful in identifying areas for improvement and clarification. We have carefully considered and addressed each of your comments, and present here our plans for revising the manuscript.

Following are a few of the concerns that require clarification.

R2.1:
1. Data collection from X (formerly Twitter) has become increasingly difficult and costly. In earlier versions of the Twitter API, researchers could easily gather data using queries and relevant keywords without limitations, which contrasts sharply with the current restrictions on the X developer platform. Including a detailed explanation of your data collection strategy, ideally in Section 2, would be highly beneficial. I understand that some data and data collection details are mentioned in section 2.4. However, including how the data was gathered will be useful. It could serve as valuable guidance for researchers working in the social media field, helping them navigate these new challenges.

We acknowledge the oversight in not providing sufficient detail about our data collection process. To address this, we will add a new subsection in Section 2 titled "2.1 Data Collection". This section will include the following information to provide clarity on the methodology:

" The specifics of data collection can be found in de Bruijn et al. (2017) and de Bruijn et al. (2019). The following section describes the processing performed by de Bruijn et al. (2017) and de Bruijn et al. (2019) followed by an overview of the additional processing performed in this study, which is described in detail in Section 2.2. The full data was collected based on the former Twitter (X) API in eleven languages (Bruijn et al. 2019). The data collection and processing involve three main types of input data. First, the authors of de Bruijn et al. (2017, 2019) used a database of known geo-locations, which contains over 4 million geographical locations including cities, towns, villages, and administrative divisions, along with alternative names and translations. Second, they collected tweets and associated metadata in real-time through the Twitter (X) streaming API using flood-related keywords in eleven languages, gathering 55.1 million tweets between July 2014 and July 2017. The keywords included terms like "flood," "flooding," and "inundation" and their equivalents in other languages. Third, they utilized GIS shapefiles of global time zones and analyzed Wikipedia articles to obtain lists of the 1000 most commonly used words per language (excluding location names with populations over 100,000). The data processing involved matching tweet text to the gazetteer through toponym recognition, scoring candidate locations based on spatial indicators, grouping related tweets, and using a voting process for toponym resolution. The system processes tweets in 24-hour windows and maintains a toponym resolution table to enable real-time geoparsing of new incoming tweets. Relevance to flooding was further ensured by classification and pre-selection based on BERT.

Based on this data we additionally performed a combination of keyword and geolocation searches during the data pre-processing to obtain tweets related to flooding events in our study areas. We analyze a sample of Twitter (X) posts (n=43,287) collected from 2014 to 2022. Our sample includes all tweets posted during this time containing one or more of the three flood-related keywords (Hochwasser, Überflutung, Flut) written in German and geotagged within Germany. The table for all keywords in other languages is available at: https://www.nature.com/articles/s41597-019-0326-9/tables/2 (de Bruijn et al., 2019)."

R2.2:

2. In Table 1, the events E1, E2, E3, E4, and E5 are well-documented. However, I am curious about how the severity or impact of the floods was measured. Was this evaluation based on factors such as the number of casualties, the extent of the affected areas, or other key indicators? Clarifying this would provide a clearer understanding of the flood classification.

We appreciate your inquiry about the flood severity classification. The classification is based on official warning levels, which are issued based on the return period. We will edit the table and related text, for the column "Flood type" to show a more direct translation of the warning levels (mean flood, large flood, extreme flood). To clarify this, we will also add the following explanation in the text after line 186:

"Flood severity was classified based on on official warning levels by LfU and LUBW (2024) using return periods of water levels. This classification corresponds to the official warning levels in Germany."

Added citation: LfU & LUBW (2024). Klassifizierung der Hochwasser-Situation am Pegel . Hochwasserzentralen.de/Info. https://www.hochwasserzentralen.de/info

3. While the observation that greater topic diversity in tweets may indicate high-impact events is intriguing (mentioned under section 3.1), I have some concerns regarding the potential for ambiguity in the analysis of specific disaster impact factors, such as casualties or damages. The detection of a higher number of topics may introduce noise and reduce clarity when focusing on key indicators of event severity. In particular, diverse topics may dilute attention on actionable information and make it more difficult to derive clear insights related to specific impact metrics. I suggest that the authors address how they mitigate this potential ambiguity when assessing diverse topics. Clarifying whether any topic weighting or hierarchies are applied could strengthen the argument and ensure that topic diversity does not introduce noise.

Topic diversity and potential ambiguity:

Your concern about potential ambiguity in analyzing disaster impact factors through diverse topics is well-founded. To address this, we would like to clarify that we did not allow for one tweet to be assigned with multiple topics and how a topic is modeled with a hierarchical modeling approach, as the weighting factors involved in that step allow mitigating the potential ambiguity resulting from the presence of multiple topics at once. We will add the following explanation to Section 3.1, L230:

"To mitigate the potential noise introduced by topic diversity, we used the HDBSCAN clustering algorithm, which contains a hierarchical topic extraction step. Primary topics, i.e., topics which occur consistently, are given higher weights in the methodology, based on their mutual reachability distance. Topics with a lower relevance, while contributing to the overall understanding of the event, were given lower weights when condensing the clusters. This approach allows us to maintain focus on critical information while still capturing the broader context of the event."

R2.4:

4. While the observation that people tend to be more proactive during low-impact flooding is an interesting insight (line 250), I have some concerns about the challenges involved in processing social media data. During disaster events, social media posts often contain typos, grammatical issues, unstructured sentences, and sometimes even lack critical information such as location details. Additionally, many users share images or videos without accompanying text, further complicating the extraction of useful insights. Given these limitations, there is a high chance of missing relevant data if basic pre-processing techniques are used. It would be valuable if the authors could clarify the specific pre-processing methods they applied to address these issues. For example, how were posts with poor grammatical structure handled? Were posts with missing location data discarded or enriched through external methods? Moreover, did the analysis account for non-textual posts, such as images or videos, that might carry significant information? Addressing these points would enhance the robustness of the findings and provide a clearer understanding of how diverse social media data was managed during different phases of the disaster event.

We appreciate your thorough consideration of the challenges in processing social media data. First, we understand that we have to more clearly explain the analyzed data type in our paper. Therefore, we will replace every instant of the "Twitter (X) data" with "textual Twitter (X) data" to make sure the reader is aware of which type of data we extracted from the platform. We only look at textual posts since this is a long-term analysis and this dataset was extracted over the years to globally map flood occurrences. Also related to your other question we see a need to improve the explanation and benefits of the chosen BERT and clustering algorithm HDBSCAN and how the clusters were extracted eliminating those posts largely affected by typos in the process. Therefore, we will include a new paragraph in Section 2.2 b. (L124) addressing all the questions listed in your comment:

"While the thorough preprocessing significantly improved our textual data quality, some unstructured, non-actionable textual Twitter(X) posts will still show up in clustering. The chosen BERT embeddings are robust when confronted with word substitution attacks (Hauser, 2021). Further, we chose HDBSCAN, which combines hierarchical clustering to avoid ambiguity and density-based methods to account for the noise in the dataset. In the first clustering step, the denser areas are separated from the surrounding points to separate areas of interest from the background noise, that is, in our case, unstructured texts. Following the separation of unstructured text and clusters of similar content discussed often, a minimum spanning tree is constructed based on a weighted graph containing the embedded textual tweets as vertices and their weighted connection based on the mutual reachability distance. Based on this we construct a hierarchy of connected components, which is then used to cut the dataset into clusters within the hierarchical structure. These steps minimize ambiguity between the clusters by using a condensed clustering tree and defining the clusters by minimum cluster size (in this case 20). Our further analysis accounts for this limitation by focusing on aggregate trends rather than individual posts."

Further, we will add the following point in the discussion after (L315) focused on how we could use the non-textual information on social media to reduce uncertainties:

"Non-textual information contained in social media posts could be included to reduce uncertainties associated with unstructured grammar or spelling mistakes, for example by comparing the image content and text content of a post"

Added citation:

Hauser, J., Meng, Z., Pascual, D., & Wattenhofer, R. (2021). Bert is robust! a case against synonym-based adversarial examples in text classification. arXiv preprint arXiv:2109.07403.

R2.5:

5. The use of SBERT in this study is impressive. However, pre-trained models like SBERT are typically trained on general-purpose datasets. Given that social media data often contains domain-specific terms and jargon, particularly in the context of disaster management, SBERT may not be fully equipped to accurately represent these specialized terms. I recommend that the authors clarify whether they fine-tuned SBERT on disaster-related data or other domain-specific corpora to enhance its performance for this particular application.

We understand the reviewer's concern with using the general-purpose model over a fine-tuned one concerning performance and topic interpretability. Here, we first want to mention that applying this framework across different contexts, areas, and circumstances is a main focus and goal for future research based on this publication, therefore we do not aim to use a model that has a very narrow range of applications (see Section 4, L290, L314, L338). This study rather is intended as a feasibility demonstration and in-depth analysis of the communication dynamics during floods in Germany, which now can be expanded to other countries. The textual Twitter data was collected in 11 countries with different country-specific jargon, thus the chosen approach makes sense conceptually in our opinion, and is, to some extent, already generalizable. This highlights the potential for future application to different contexts.

Nevertheless, we tested different transformer models, which are more specific to our data alongside the one presented in the paper, including GermanBERT (https://huggingface.co/dbmdz/bert-base-german-uncased) and TwHIN-BERT (https://huggingface.co/Twitter/twhin-bert-base). Here, we compared the topic similarity before optimizing the clustering hyperparameters based on a specific transformer model and found that the topics extracted with the multilingual transformer were more distinct (less similar), which we evaluated based on the Coherence Score and Density-based Clustering Validation score. Therefore, we decided that the expected performance for a more general model is higher and we did not look into further fine-tuning.

We will add the following clarification to our methodology section:

"While SBERT is pre-trained on general-purpose datasets, we found its performance on our disaster-related corpus to be robust. For verification, we conducted an experiment where we compared a German model (GermanBERT) (Darji, 2023) and a model trained on Tweets specifically (TwHIN-BERT) (Zhang, 2023), where we found that the topics were less distinct and interpretable. Thus, we proceeded with the pre-trained SBERT model to maintain generalizability, aiming to demonstrate an approach which, in the future, can be adapted to different contexts and case studies."

Added citations:

Zhang, X., Malkov, Y., Florez, O., Park, S., McWilliams, B., Han, J., & El-Kishky, A. (2023, August). Twin-bert: A socially-enriched pre-trained language model for multilingual tweet representations at twitter. In Proceedings of the 29th ACM SIGKDD conference on knowledge discovery and data mining (pp. 5597-5607).

Darji, H., Mitrović, J., & Granitzer, M. (2023). German BERT model for legal named entity recognition. arXiv preprint arXiv:2303.05388.

R2.6:

6. While the use of SBERT and HDBSCAN for topic detection is appropriate, the paper does not provide any evaluation metrics to assess the performance of the clustering model. Including results from common clustering evaluation metrics, such as Silhouette Score, or others, would give a clearer indication of the quality of the identified clusters. I recommend that the authors include these metrics to validate the effectiveness of their clustering approach. I believe that some details might be available in supplementary

materials as mentioned in line 125, but the links to these materials cannot be found in the document.

We acknowledge the oversight in not providing clustering evaluation metrics. The silhouette coefficient is not suitable for density HDBSCAN because of the initial step, where the low-density areas are summarized as a cluster and the silhouette score is calculated based on the distance between points without taking the density into account. Therefore, we calculated the Density Based Clustering Validation Score, which is suitable for density-based clustering, and included it in the updated manuscript as follows:

"To evaluate the quality of our HDBSCAN clustering, we calculate the Density-Based Clustering Validation (DBCV) score (Moulavi, 2014). Based on the definition that clusters represent areas of higher density amongst regions of lower density, a relative validity measure is calculated by combining the shape and density properties of the cluster. The density is evaluated relative to the density in a cluster representing the background noise. We obtained a positive score of 0.24 within the DBCV range of -1 to 1, which validates the effectiveness of our clustering approach in identifying distinct topics within the tweet corpus."

Added citation:

Moulavi, D., Jaskowiak, P. A., Campello, R. J., Zimek, A., & Sander, J. (2014, April). Density-based clustering validation. In Proceedings of the 2014 SIAM international conference on data mining (pp. 839-847). Society for Industrial and Applied Mathematics.

We will also ensure that all Supporting Information are properly linked in the revised manuscript.

R2.7:

7. One of the main limitations of using social media data is its reliability and credibility, particularly in disaster management scenarios where the stakes are high. The potential for false information or rumors can undermine the effectiveness of response efforts. Therefore, it would be beneficial if the results of the model were tested against specific disaster events and compared with official records for validation. Such an approach would enhance the credibility of the findings and ensure that the insights derived from social media are both actionable and trustworthy.

The reviewer points out, that Social Media Data is a potentially unreliable source of information and may contain false information or rumors.

In response to comment R2.1, we suggested adding a section clarifying the data-gathering strategy, which also describes multiple steps to ensure the robustness of our results. Here we would also like to point the reviewer to Figure 3 from de Bruijn, 2019, which shows the validation of the textual Twitter (X) dataset used in this study based on the NatCatSERVICE data provided by Munich Re.

[Figure]

**Fig. 3** Detected events and number of "relevant" tweets. The panels show detected events globally (**a**), in the United States (**b**), in Indonesia (**c**) and in Tanzania (**d**). Events in admin1 regions are included. This figure only shows tweets (in blue) when part of a detected event. A selection of events is labelled for reference. *The number of tweets is displayed as a 3-day average as otherwise the bars become too small to be visible in the figure.

Additionally, users hosted by official organizations, such as @hochwasserportal_de are explicitly mentioned in the text to show that those topics are not linked to rumors or false information.

We addressed the limitation pointed out by the reviewer in the discussion section and suggested that multiple platforms could be used comparatively to detect content that is falsely published on one platform. This would further improve reliability.

Line 308-312:" Furthermore, social media platforms like Twitter (X) represent only a subset of society. Consequently, insights drawn from Twitter (X) data may not fully capture the different experiences, perspectives, and actions present within the broader population. This limitation can be addressed by evaluating the topics discussed on alternative social media platforms other than Twitter (X), which also ensures robustness towards the fluctuations of access to data from individual providers."

We will add an additional sentence after these lines to clarify further uncertainties:

"In de Bruijn, 2019, Figure 3 the textual Twitter (X) data is successfully validated against NatCatSERVICE data provided by Munich Re, further ensuring robustness."

Minor typos

1. Line number 60 - Reference is not given within brackets; Konya and Nematzadeh (2024).

2. Line number 98 - I assume the usage of the word "The" came in twice was a mistake.

3. While including in-text citations with multiple references (such as in line 15), the order of reference can be year-wise. This improves readability.

We appreciate your attention to detail. We have made the following corrections:

1. Line 60: The citation will read (Konya and Nematzadeh, 2024) in the revised manuscript.
2. Line 98: We will remove the duplicate "The" from the sentence.
3. We will reorder all multiple in-text citations chronologically to improve readability.